# Water cluster in hydrophobic crystalline porous covalent organic frameworks

Ke Tian Tan [1], Shanshan Tao[1], Ning Huang [1] & Donglin Jiang [1✉]

Progress over the past decades in water confinement has generated a variety of polymers and porous materials. However, most studies are based on a preconception that small hydrophobic pores eventually repulse water molecules, which precludes the exploration of hydrophobic microporous materials for water confinement. Here, we demonstrate water confinement across hydrophobic microporous channels in crystalline covalent organic frameworks. The frameworks are designed to constitute dense, aligned and one-dimensional polygonal channels that are open and accessible to water molecules. The hydrophobic microporous frameworks achieve full occupation of pores by water via synergistic nucleation and capillary condensation and deliver quick water exchange at low pressures. Water confinement experiments with large-pore frameworks pinpoint thresholds of pore size where confinement becomes dominated by high uptake pressure and large exchange hysteresis. Our results reveal a platform based on microporous hydrophobic covalent organic frameworks for water confinement.

---

[1] Department of Chemistry, Faculty of Science, National University of Singapore, 3, Science Drive 3, Singapore 117543, Singapore. ✉email: chmjd@nus.edu.sg

Water confinement is ubiquitous and it plays vital roles in geological, biological and artificial systems with many applications[1–8]. To develop water confinement systems, extensive studies have been conducted over the past 60 years on a variety of hydrogel polymers with various hydrophilic backbones and different functionalities[1–3]. However, the lack of mechanistic strength and immobile water state have precluded their real implementation in water uptake and transport. To create porous structures, researchers have relied on bottom-up strategy via self-assembly of block polymers[9–11]. However, it is difficult to prepare stable and well-defined channels from phase-separated polymers. In recent years, a variety of porous polymers bearing intrinsic porosity has been studied extensively for water confinement and uptake[4–6]. Nevertheless, the design of hydrophobic small-pore materials is precluded by a preconception that hydrophobic small pores would eventually repulse water molecules and cannot enable water confinement and uptake.

We anticipate hydrophobic supermicroporous covalent organic frameworks (COFs) with compact pore void is indispensable for water cluster occupation. COFs as a crystalline porous framework material, can be constructed via bottom-up polymerization through topological design. Upon judicious selection of building blocks, aromatic components can connect via covalent bonds to grant extended crystalline structure and hydrolytic stability in COFs. Particularly, simplistic design can generate permanent supermicroporous voids that are implausible to collapse. These distinct features can be combined in COFs and are highly desired for supermicropores but hardly accessible to other porous analogues. In this work, we constitute hydrophobic microenvironment entwined with "pseudo-hydrophilicity" strips to promote water confinement at low pressure. We further dive in to investigate the impact of pore size and shape to pinpoint the size threshold and shape for favourable water molecules confinement. We unexpectedly found that pore size threshold is distinctive to each pore shape in creating similar pore environment. We establish the size borderline of trigonal, hexagonal and tetragonal frameworks to eliminate the hysteretic behaviour in water uptake.

## Results

**Design and crystal structures of trigonal COFs.** In this study, we demonstrate water confinement across hydrophobic microporous channels in crystalline and porous covalent organic frameworks (COFs). COFs are a unique class of polymers that enable the creation of dense, aligned and one-dimensional (1D) channels, with a diversity of applications[12,13]. The targeted COFs were constructed to possess predesigned 1D channels with different polygonal shape, pore size and wall environment. We synthesized microporous trigonal hydrophobic COFs, i.e., HFPTP-PDA-COF[14], HFPTP-DMePDA-COF and HFPTP-BPDA-COF[15] via condensation of 2,3,6,7,10,11-hexakis(4-formylphenyl) triphenylene (HFPTP) with 1,4-phenylenediamine (PDA), 2,5-dimethyl-1,4-phenylenediamine (DMePDA) and 1,1'-biphenyl-4,4'-diamine (BPDA) under solvothermal conditions as previously reported (Fig. 1). Their chemical structures were characterised by analytic methods (Supplementary Figs. 1 and 2).

Powder X-ray diffraction (PXRD) pattern of the newly synthesized HFPTP-DMePDA-COF revealed peaks at 4.03°, 6.97°, 8.12°, 10.62°, 11.07° and 23.22°, which were assigned to the (100), (110), (200), (210), (300) and (001) facets, respectively (Fig. 2a, black curve). We used density-functional based tight-binding (DFTB+) calculations to optimise the conformation of single 2D polymer sheet and its configuration of stacking modes. The AA-stacking mode (Fig. 2a, red curve) reproduces the PXRD peak position and intensity. The HFPTP-DMePDA-COF adopts a $P_3$ space group of with unit-cell parameters of $a = b = 25.8092$ Å,

$c = 4.5279$ Å, $\alpha = \beta = 90°$ and $\gamma = 120°$ (for atomistic coordinates, see Supplementary Table 1). We conducted Pawley refinements (Fig. 2a, green curve) and confirmed the correctness of the PXRD peak assignments, as indicated by a negligible difference (Fig. 2a, yellow curve) with $R_{wp} = 3.72\%$ and $R_p = 2.15\%$. Rietveld refinement also yields a $P_3$ space group. The presence of (001) facet at 23.22° (Fig. 2a, inset) suggests an extended structural ordering with an interlayer interval of 3.83 Å along the $z$ direction. As the trigonal topology offers the smallest pores among COFs, HFPTP-DMePDA-COF consists of dense, aligned and microporous triangular 1D channels (Fig. 1c). On the other hand, HFPTP-PDA-COF (Fig. 2b, red curve) and HFPTP-BPDA-COF (Fig. 2b, blue curve) displayed the same PXRD patterns as reported[14,15], adopting AA stacking structures to form aligned microporous 1D channels (Fig. 1a, b and d). With the AA stacking structure, the trigonal COFs develop hydrophobic microporous channels in which dense C–H sites are extruded from each pore wall to form an extended inner 'carpet' while the C = N linkages are polarisable to spot "hydrophilic" sites owing to their partially charged dipole moments.

**Porosity of trigonal COFs.** These trigonal COFs exhibited typical type I nitrogen sorption isotherms, manifesting their microporous features (Fig. 2c, e and g). The HFPTP-PDA-COF and HFPTP-DMePDA-COF exhibited a Brunauer-Emmett-Teller (BET) surface area of 662 and 480 $m^2g^{-1}$, a pore volume of 0.32 and 0.24 $cm^3g^{-1}$, respectively (Fig. 2d and f, Supplementary Table 2). They exhibited pore sizes of 1.1 and 1.4 nm (Fig. 2d and f), which are consistent with their crystal structures. On the other hand, HFPTP-BPDA-COF exhibited a BET surface area of 758 $m^2 g^{-1}$, a pore volume of 0.39 $cm^3 g^{-1}$ and pore sizes of 1.2 and 1.6 nm (Fig. 2g and h). Noticeably, the cumulative pore size distribution profiles revealed that the contribution of 1.1- or 1.2-nm pore is significantly larger than that of 1.4- or 1.6-nm pore, indicating that the small-size supermicropore predominates the materials (Fig. 2d, f and h). Compared to HFPTP-PDA-COF and HFPTP-DMePDA-COF, HFPTP-BPDA-COF has a 0.1-nm larger pore size. Minor large pores at 1.8, 4.1 and 3.8 nm are observed in HFPTP-PDA-COF, HFPTP-DMePDA-COF and HFPTP-BPDA-COF due to an incomplete connection between neighbouring pores.[15] Nevertheless, these pores occupy a negligible percentage in terms of pore volume and thus do not impact the water sorption behaviour.

**Water sorption and confinement of trigonal COFs.** The water vapour sorption isotherms of HFPTP-PDA-COF (Fig. 3a–c, red dots and curves) resemble a typical $S$-shaped type-V sorption curve, which is characteristic of hydrophobic materials[16–18]. The HFPTP-PDA-COF at 25 °C (Fig. 3a, red dots and curve) exhibited an induction pressure zone (P/P$_0$) of 0–0.4, where water molecules are nucleated in the small channel. After the induction zone, a steep water uptake occurs at a low P/P$_0$ of 0.4 and completes at P/P$_0$ of 0.41. The quick uptake originates from capillary condensation of water molecules in the microporous 1D channels. Similarly, HFPTP-PDA-COF at 15 °C and 10 °C (Fig. 3b and c, red dots and curve) displayed an induction P/P$_0$ zone of 0–0.37, 0–0.38 and a sharp capillary condensation at P/P$_0$ of 0.37–0.40, 0.38–0.40. Such a sharp uptake at a low pressure is attractive for water confinement. In contrast, HFPTP-DMePDA-COF with the same pore size but bearing methyl groups on pore walls totally changes the behaviour. The HFPTP-DMePDA-COF exhibited an induction P/P$_0$ zone of 0–0.65 at 25 °C (Fig. 3a, blue dots and curve), 0–0.63 at 15 °C (Fig. 3b, blue dots and curve) and 0–0.61 at 10 °C (Fig. 3c, blue dots and curve), which are much far broader than those of HFPTP-PDA-COF. More significantly,

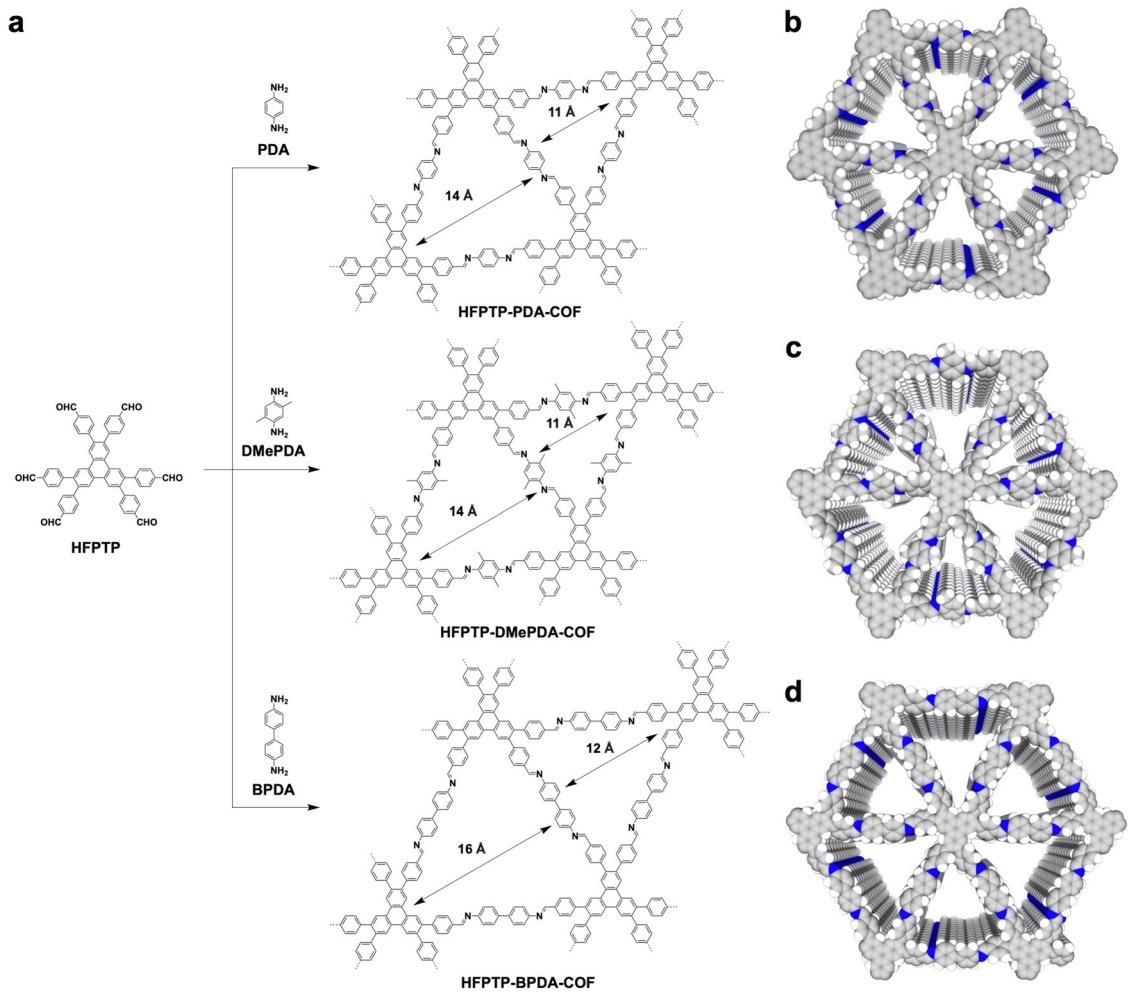

**Fig. 1 Hydrophobic supermicroporous 1D channels in trigonal COFs. a** Schematic of synthesis of HFPTP-PDA-COF, HFPTP-DMePDA-COF and HFPTP-BPDA-COF. Reconstructed structures of (**b**) HFPTP-PDA-COF, (**c**) HFPTP-DMePDA-COF and (**d**) HFPTP-BPDA-COF.

HFPTP-DMePDA-COF did not show a steep uptake but displayed a rather sluggish adsorption over a higher and broader P/$P_0$ range of 0.6–0.9 at 25 °C, 0.6–0.9 at 15 °C and 0.6–0.8 at 10 °C, respectively. These drastic changes reflect that a small perturbation of the pore walls completely alters the water confinement in the hydrophobic triangular supermicropores.

The HFPTP-BPDA-COF with a pore size of 1.2 nm exhibited an induction P/$P_0$ zone of 0–0.52 at 25 °C (Fig. 3a, black dots and curve), 0–0.51 at 15 °C (Fig. 3b, black trigons and curve) and 0–0.50 at 10 °C (Fig. 3c, black squares and curve), respectively, which are much broader than those of HFPTP-PDA-COF with a pore size of 1.1 nm (Fig. 3a–c, red symbols and curves). Capillary condensation was observed but it occurred at much higher pressures of 0.52–0.58 at 25 °C, 0.52–0.57 at 15 °C and 0.50–0.58 at 10 °C, respectively. The greatly broadened induction zone together with a dramatic shift to higher pressure reflects that water is also highly sensitive to the pore size of triangular channels, where large-pore trigonal COFs with a pore size larger than 1.2 nm are no longer favourable for water confinement.

We evaluated the pore occupancy by assuming the density of water to be 0.997 g cm$^{-3}$ using Eq. (1). Notably, HFPTP-PDA-COF and HFPTP-BPDA-COF at 25 °C exhibited a pore occupancy of 93% and 92%, respectively, denoting that the channel void space is highly accessible to water molecules (Fig. 3d). In contrast, HFPTP-DMePDA-COF at 25 °C displayed a pore occupancy of only 49%, which is only half of those of HFPTP-PDA-COF and HFPTP-BPDA-COF. The much lower

pore accessibility originates from the methyl units on the wall surface which impose a severe steric hindrance for water access to the neighbouring hydrophilic N chains. This spatial restriction extends along the z direction on each pore walls and causes difficulty of water cluster formation to reach a high pore filling.

$$Pore\ occupancy = \frac{Uptake\ capacity}{Pore\ volume} \times 100\% \qquad (1)$$

**Regime of hexagonal and tetragonal COFs**. With the distinct pore size and environment thresholds in trigonal COFs, we further investigated if pore shape would trigger different confinement property (Fig. 4a–h, Supplementary Table 3). COFs are unique to enable the predesign of 1D channels with different polygonal shapes and sizes while retaining the same pore wall structures; we thus synthesised microporous tetragonal TFBCz-PDA-COF[19] (Fig. 4a, pore size = 1.5 nm, Supplementary Fig. 3a–c), microporous hexagonal TTA-TFB-COF[20] (Fig. 4b, pore size = 1.5 nm, Supplementary Fig. 3d–f), mesoporous tetragonal TFPPy-PDA-COF[21] (Fig. 4c, pore size = 2.1 nm, Supplementary Fig. 4a–c) and mesoporous hexagonal TPB-DMTP-COF[22] (Fig. 4d, pore size = 3.2 nm, Supplementary Fig. 4d–f). These COFs create dense and aligned 1D open channels, similar to the trigonal COFs in terms of hydrophobic C–H "carpet" and "hydrophilic" C=N sites on walls (Fig. 4e–h).

Microporous tetragonal TFBCz-PDA-COF (Supplementary Fig. 5a and c) and hexagonal TTA-TFB-COF (Supplementary Fig. 5a and d)

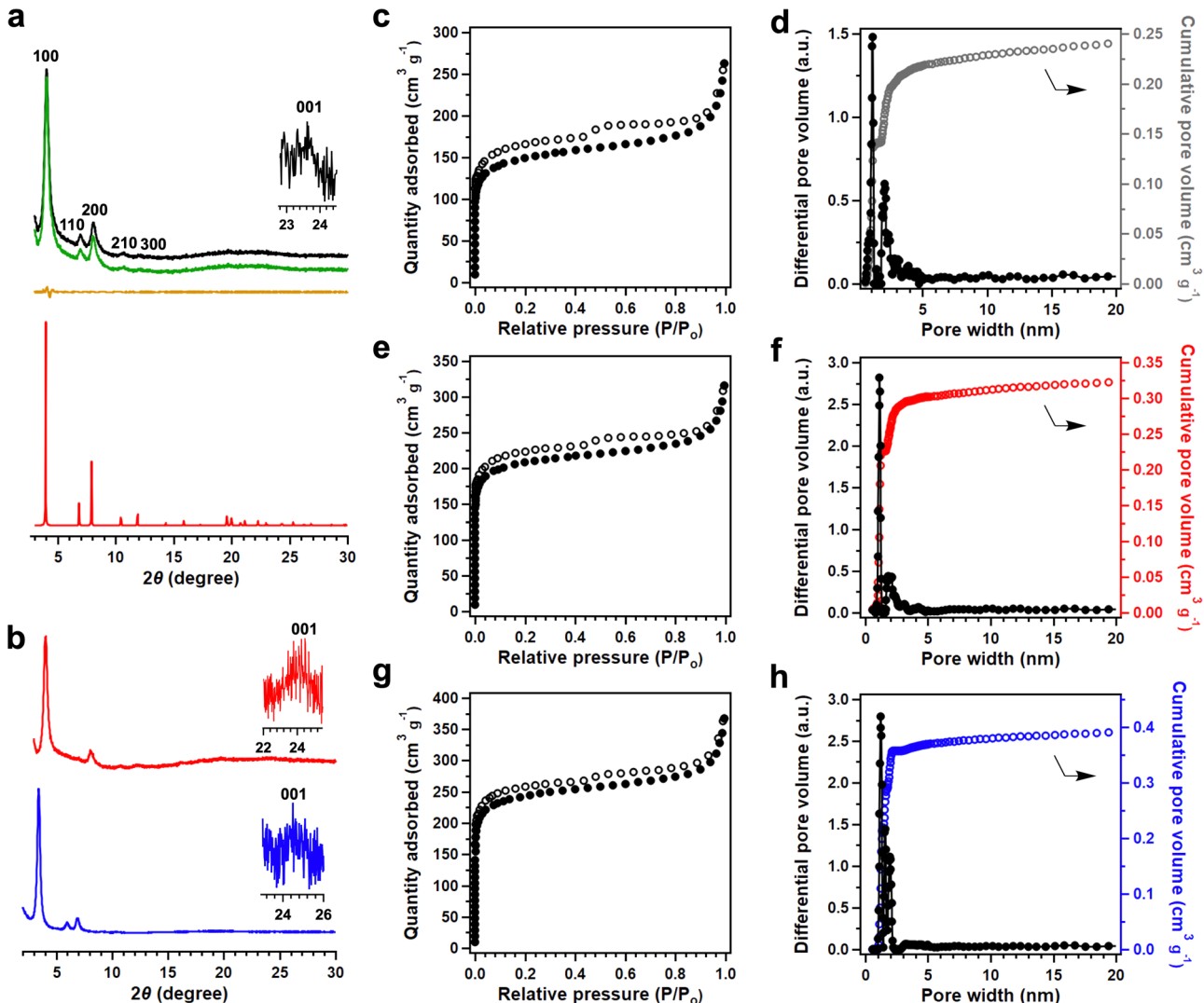

**Fig. 2 PXRD and Porosity. a** PXRD patterns of HFPTP-DMePDA-COF (black curve), the Pawley refinement result (green curve) and their difference (yellow curve) and the AA-stacking mode of the $P_3$ space group (red curve). **b** PXRD patterns of HFPTP-PDA-COF (red curve) and HFPTP-BPDA-COF (blue curve). **c**, Nitrogen sorption isotherm of HFPTP-DMePDA-COF. **d** Pore size distribution and pore volume of HFPTP-DMePDA-COF. **e** Nitrogen sorption isotherm of HFPTP-PDA-COF. **f**, Pore size distribution and pore volume of HFPTP-PDA-COF. **g** Nitrogen sorption isotherm of HFPTP-BPDA-COF. **h** Pore size distribution and pore volume of HFPTP-BPDA-COF.

have a pore size of 1.5 nm and achieve similar induction $P/P_0$ zone of 0–0.44 at 25 °C (Fig. 4i, red and blue symbols and curves), 0–0.42 at 15 °C (Fig. 4j, red and blue symbols and curves) and 0–0.42 at 10 °C (Fig. 4k, red and blue symbols and curves), respectively. Both COFs exhibited a sharp uptake at $P/P_0$ of 0.4–0.45 owing to capillary condensation (Fig. 4i–k, red and blue symbols and curves). Noticeably, TFBCz-PDA-COF and TTA-TFB-COF at 25 °C exhibited a pore occupancy of 79% and 97%, respectively (Fig. 4l). Although tetragonal TFBCz-PDA-COF and hexagonal TTA-TFB-COF have a larger micropore (1.5 nm) than the trigonal HFPTP-PDA-COF (1.1 nm), however, their induction zones and capillary condensation pressures are close to those of trigonal HFPTP-PDA-COF. These results revealed that the channel shape plays an important role in water confinement and tetragonal and hexagonal COFs shift the threshold of pore size to a large micropore side. A distinct feature of water exchange in these hydrophobic micropores (adsorption and desorption) is that it occurs at a low $P/P_0$ around 0.4 within a narrow pressure window (small hysteresis loop), which is attractive for water flux and heat-pump implementation[23].

Compared to microporous analogues, mesoporous tetragonal and hexagonal COFs greatly increase the uptake pressure (Fig. 4i–k, green and black symbols and curves). In the case of TFPPy-PDA-COF (pore size = 2.1 nm), the induction $P/P_0$ zone was greatly broadened to 0 – 0.56 at 25 °C (Fig. 4i, green dots and curve), 0–0.55 at 15 °C (Fig. 4j, green trigons and curve) and 0–0.58 at 10 °C (Fig. 4k, green squares and curve). A more explicit broadening was observed for the hexagonal large-pore TPB-DMTP-COF (pore size = 3.2 nm), which exhibited an induction $P/P_0$ zone of 0–0.72 at 25 °C (Fig. 4i, black dots and curve), 0–0.72 at 15 °C (Fig. 4j, black trigons and curve) and 0–0.68 at 10 °C (Fig. 4k, black squares and curve), respectively. A further drawback is that the mesoporous COFs display a large hysteresis loop to complete the adsorption–desorption exchange cycle (Fig. 4i–k, black and green symbols and curves) similar to mesoporous silica FSM-16, MPS-1, MPS-2 and MPS-3[24,25]. For example, TPB-DMTP-COF exhibited a hysteric loop over a wide $P/P_0$ range of 0.48–0.9 at 25 °C, 0.44–0.9 at 15 °C and 0.42–0.9 at 10 °C, respectively. The 2.2-nm pore TFPPy-PDA-COF at 25 °C

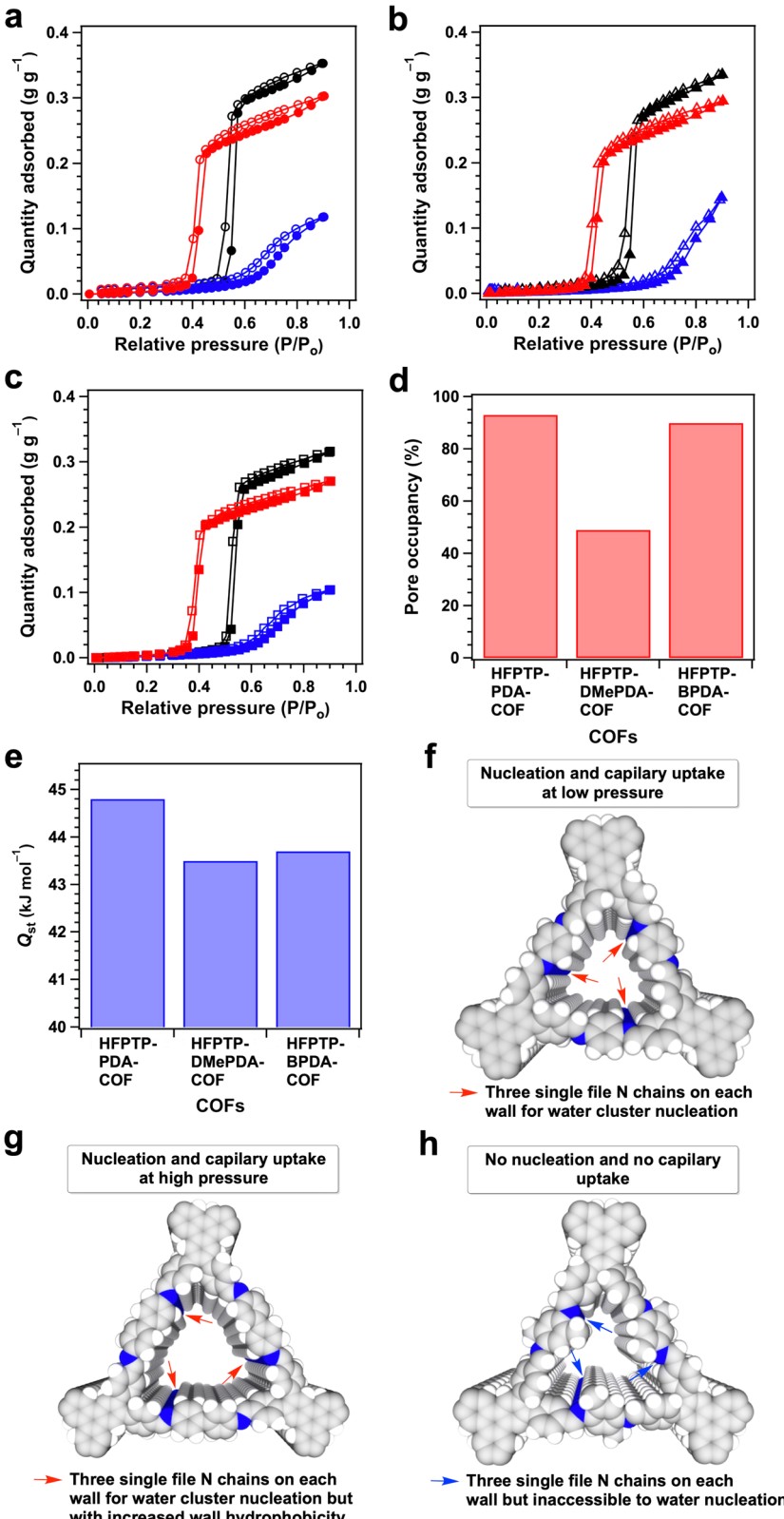

**Fig. 3 Water vapour uptake and pore accessibility in trigonal COFs. a** Vapour sorption isotherms of HFPTP-PDA-COF (red), HFPTP-DMePDA-COF (blue) and HFPTP-BPDA-COF (black) measured at 25 °C (filled circle–adsorption branch, empty circle–desorption branch). **b** Vapour sorption isotherms of HFPTP-PDA-COF (red), HFPTP-DMePDA-COF (blue) and HFPTP-BPDA-COF (black) measured at 15 °C (filled trigon – adsorption branch, empty trigon – desorption branch). **c** Vapour sorption isotherms of HFPTP-PDA-COF (red), HFPTP-DMePDA-COF (blue) and HFPTP-BPDA-COF (black) measured at 10 °C (filled square – adsorption branch, empty square – desorption branch). **d** Pore occupancies of trigonal COFs. **e** Isosteric heat of adsorption ($Q_{st}$) of trigonal COFs. **f–h** The hydrophobic C–H 'carpet' and 'hydrophilic' single file C=N chains (blue chains) on walls of (f) HFPTP-PDA-COF, (g) HFPTP-BPDA-COF and (h) HFPTP-DMePDA-COF.

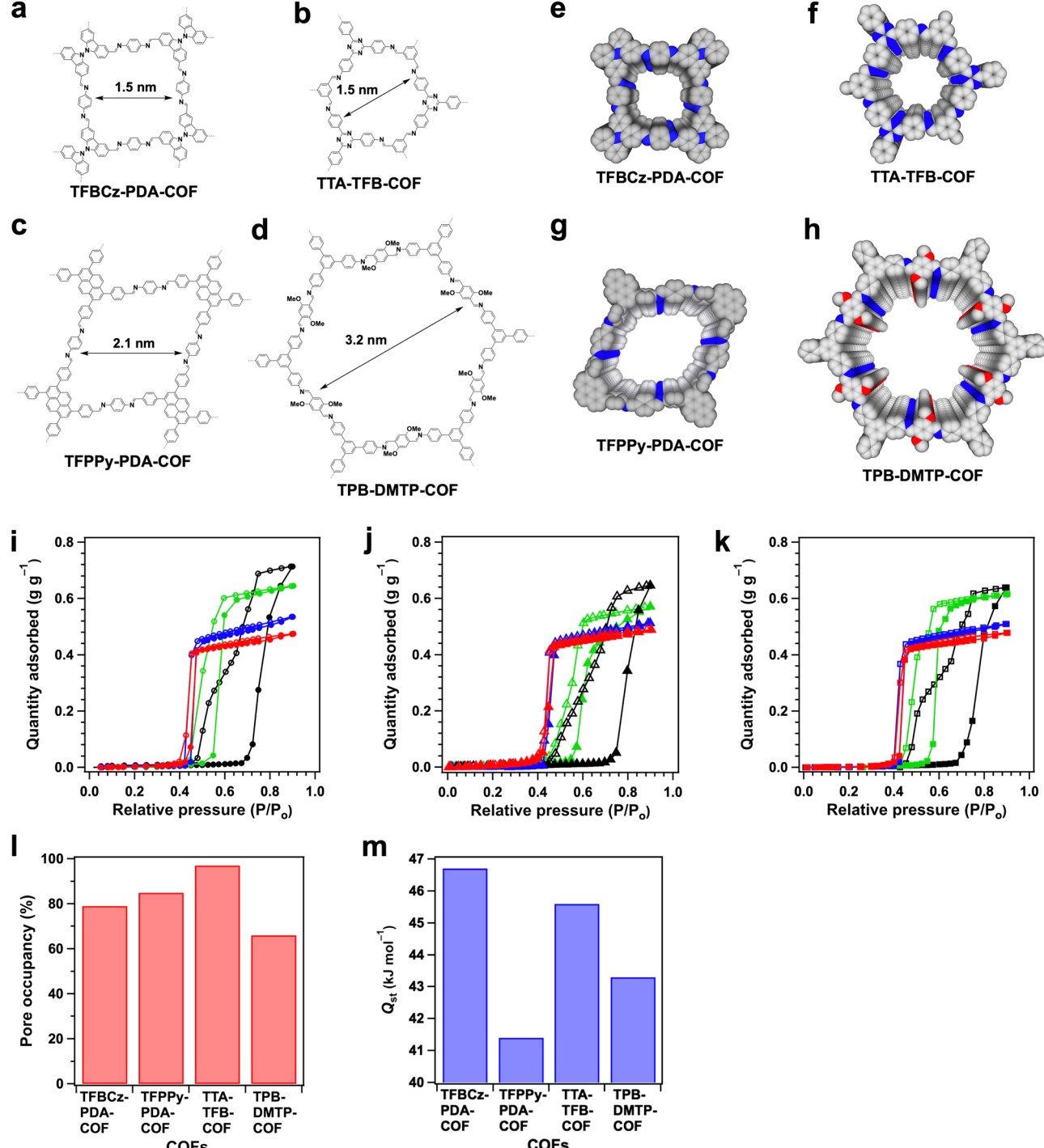

**Fig. 4 Diversity of pore shape and size of hydrophobic 1D channels. a** Topological synthesis of TFBCz-PDA-COF. **b** Topological synthesis of TTA-TFB-COF. **c** Topological synthesis of TFPPy-PDA-COF. **d** Topological synthesis of TPB-DMTP-COF. Reconstructed structures of (**e**) TFBCz-PDA-COF, (**f**) TTA-TFB-COF, (**g**) TFPPy-PDA-COF and (**h**) TPB-DMTP-COF. **i** Vapour sorption isotherms of TFBCz-PDA-COF (red), TTA-TFB-COF (blue), TFPPy-PDA-COF (green) and TPB-DMTP-COF (black) measured at 25 °C (filled circle – adsorption branch, empty circle – desorption branch). **j** Vapour sorption isotherms of TFBCz-PDA-COF (red), TTA-TFB-COF (blue), TFPPy-PDA-COF (green) and TPB-DMTP-COF (black) measured at 15 °C (filled trigon – adsorption branch, empty trigon – desorption branch). **k** Vapour sorption isotherms of TFBCz-PDA-COF (red), TTA-TFB-COF (blue), TFPPy-PDA-COF (green) and TPB-DMTP-COF (black) measured at 10 °C (filled square – adsorption branch, empty square – desorption branch). **l** Pore occupancy. **m** $Q_{st}$ of TFBCz-PDA-COF, TFPPy-PDA-COF, TTA-TFB-COF and TPB-DMTP-COF.

displayed a pore occupancy of 85% while the 3.1-nm pore TPB-DMTP-COF dropped the pore occupancy to only 57% (Fig. 4l).

To demonstrate the importance of being 1D channels, we synthesised low-crystallinity TTA-TFB-CMP (Supplementary Fig. 6) which possesses the same components as TTA-TFB-COF. Surprisingly, TTA-TFB-CMP diminishes the steep uptake and portrays only a sluggish uptake (Supplementary Fig. 6d). This result indicates that the 1D channel is of the utmost importance for water confinement.

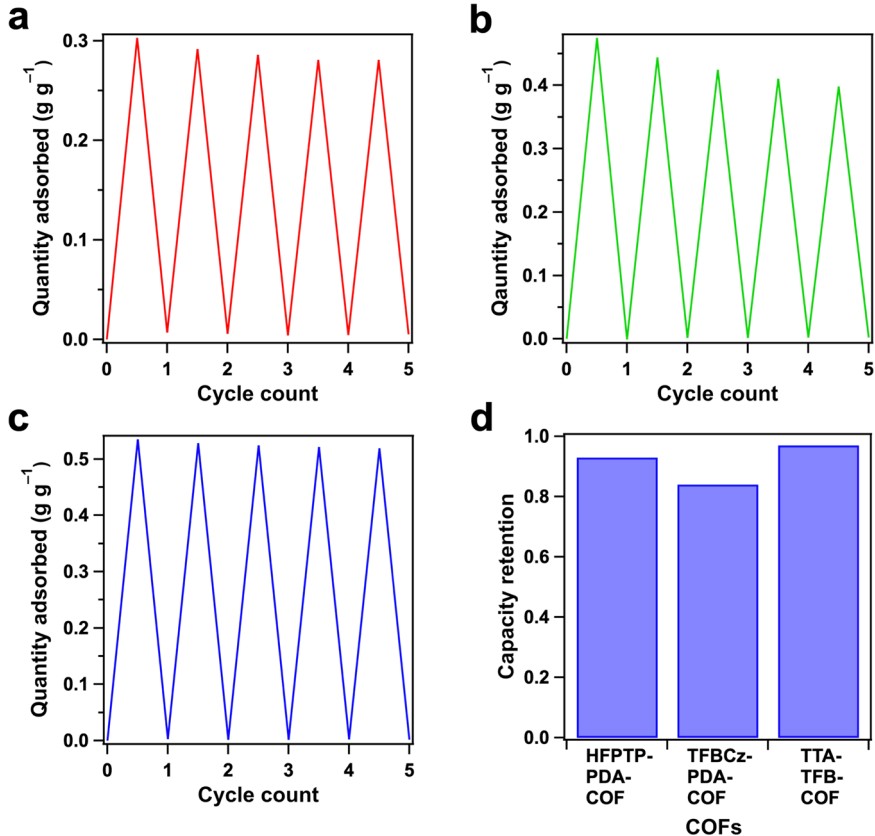

**Fig. 5 Cycle performance of water confinement. a** Cycle performance of HFPTP-PDA-COF. **b** Cycle performance of TFBCz-PDA-COF. **c** Cycle performance of TTA-TFB-COF. **d** Capacity retention of HFPTP-PDA-COF, TFBCz-PDA-COF and TTA-TFB-COF.

Bulk water[26] possesses heat of vaporisation of 44 kJ mol$^{-1}$ at 25 °C. We investigated the isosteric heat of adsorption ($Q_{st}$) of water confinement in COFs (Supplementary Fig. 7). The HFPTP-PDA-COF exhibited a $Q_{st}$ value of 44.8 kJ mol$^{-1}$, while HFPTP-DMePDA-COF and HFPTP-BPDA-COF displayed a $Q_{st}$ value of 43.5 and 43.7 kJ mol$^{-1}$, respectively (Fig. 3e, Supplementary Table 3). The $Q_{st}$ value (Fig. 4m) of mesoporous TFPPy-PDA-COF and TPB-DMTP-COF is 41.4 and 43.3 kJ mol$^{-1}$, respectively, which are lower than those of supermicroporous TFBCz-PDA-COF (46.7 kJ mol$^{-1}$) and TTA-TFB-COF (45.6 kJ mol$^{-1}$). These results manifest that hydrophobic microporous COFs enable stronger water confinement than mesoporous COFs (Supplementary Table 3), highlighting their high-water capture ability. By merging the feature of narrow induction pressure zone, steep uptake and high absorption heat into one material, it is safe to reach a conclusion that hydrophobic microporous channels are superb for water confinement and exchange.

## Discussion

The above studies on COFs pinpoint a clear threshold of pore size, shape and environment for water confinement and exchange. In trigonal supermicroporous COFs, pore walls are densely covered with aromatic C–H 'carpet', endowing the porous space with a high hydrophobic feature where water molecules are expected to interact with the walls weakly via van der Waals forces (Fig. 3f–h). On the other hand, the C=N linkage site is polarised by the large electronegativity of nitrogen atom to form partially charged species ($C^{\delta+}$–$N^{\delta-}$) and trigger dipolar 'hydrophilicity'. Noticeably, the polarised C=N site is aligned vertically to form single file $C^{\delta+}$–$N^{\delta-}$ chains, constituting a strip of 'hydrophilic' zone along the $z$ direction on each wall (Fig. 3f–h). These 'hydrophilic' strips on the hydrophobic walls offer sites for nucleation of water clusters in the channels. Such an aligned yet bicontinuous hydrophobic and 'hydrophilic' pore interface is unique in COFs and inaccessible to other porous polymers. In HFPTP-PDA-COF, at a low relative pressure ($P/P_0 = 0$–0.38), the 'hydrophilic' C=N strip serves as an anchor to enable the nucleation of water clusters (Fig. 3f). As the pressure passes this threshold, the water clusters grow sharply until 0.40 $P/P_0$ and instant pore filling occurs via capillary condensation[27]. In contrast, HFPTP-DMePDA-COF is different in the confinement process and mechanism. The methyl groups owing to hyperconjugation effect weaken polarisation of the C=N linkages[28] to decrease the 'hydrophilicity' and spatially set a steric barrier to prevent the access to water molecules (Fig. 3h). These two-fold effects broaden the induction pressure zone and eliminate capillary condensation. In this case, water molecules are confined by sluggish diffusion via water-water intermolecular hydrogen bonds, resulting in an isotherm convex to the pressure axis. The 1.2-nm pore HFPTP-BPDA-COF with biphenyl linker constitutes much dense hydrophobic C–H sites on the pore wall while the C=N sites retain almost unchanged in density and are accessible to water molecules (Fig. 3g). As a result, HFPTP-PDA-COF with a 0.1-nm increased pore size greatly shifts the confinement to a much higher pressure. In the tetragonal and hexagonal microporous COFs, their pore wall environments are similar to HFPTP-PDA-COF, so that nucleation and capillary condensation occur at a low pressure. In contrast, in mesoporous tetragonal and hexagonal COFs, the wall environments become similar to HFPTP-BPDA-COF while the increased pore size likely triggers further shift to high pressure and large hysteresis loop.

The low pressure, high pore occupancy and quick water exchange cycle are attractive in heat pump implementation. We investigated cycling stability of HFPTP-PDA-COF (Fig. 5a, Supplementary Fig. 5b), TFBCz-PDA-COF (Fig. 5b,

Supplementary Fig. 5c) and TTA-TFB-COF (Fig. 5c, Supplementary Fig. 5d). They can all retain more than 84% and even up to 97% of their maximum capacity after 5 cycles (Fig. 5d). Robust capacity retention highlights the superiority of these COFs as heat-pump adsorbents for pro-longed cycling. Noticeably, all the microporous trigonal, tetragonal and hexagonal COFs retain their structural integrity with the identical peaks in PXRD patterns (Supplementary Figs. 8–10) and FT IR spectra (Supplementary Figs. 2, 11 and 12). The robustness against water vapour enables COFs as a suitable candidate for adsorption energy recovery and desorption cooling systems which require porous materials with high water capacity and stability[16]. This is superior to some metal-organic frameworks that are prone to structural degradation after continuous exposure to water[29]. The HFPTP-PDA-COF after 5 cycles exhibited a BET surface area of 505 m$^2$ g$^{-1}$ and a pore volume of 0.23 cm$^3$ g$^{-1}$ (Supplementary Fig. 13). Noticeably, the 1.1–nm micropore is well retained while defects are self-healed to define a flat plateau in the pore size distribution profile after cycle use (Supplementary Fig. 13c, red curve).

By developing a series of crystalline porous covalent organic frameworks, we have successfully shown water confinement in hydrophobic microporous channels. Microporous COFs are unique in that they merge high accessibility, steep uptake at low pressure, negligible hysteresis loop, large adsorption heat and robust cyclability in one material. This feature originates from the twofold effects of small pores: one is that the pore walls are covered by dense aromatic C–H 'carpet' which coat a hydrophobic inner wall; another one is that the built-in C=N linkages polarise to afford 'hydrophilicity' of certain degree, so that their vertical alignment along the z direction forms single file C=N chain to develop a strip to enable the nucleation of water cluster. Such an ordered yet spatially separated bicontinuous structure on each pore wall renders the microporous channels able to confine water clusters quickly.

Among the microporous COFs, trigonal channel is the most promising one as it is much more sensitive to pore size and environment. A slight increase in pore size by 0.1 nm drastically changes the confinement while a small wall perturbation with methyl unit switches off the capillary condensation to a sluggish diffusion mechanism. In contrast, mesopores COFs (pore size > 2 nm) result in high uptake pressure, large hysteresis, limited pore occupancy and low adsorption heat – these features lead to an unambiguous conclusion that large hydrophobic pores should be precluded for water confinement, uptake and exchange. COFs are unique in that their skeletons and pores are both predesignable. The present approach is not limited to the current skeletons but widely extendable to other microporous COFs with different building blocks, linkages and pore sizes to achieve water confinement. These COFs would be applicable in many areas, including water uptake, transport and separation as well as heat-pump energy conversion.

## Methods

**Characterisations**. Powder X-Ray diffraction (PXRD) patterns were recorded on Bruker D8 Advance by depositing powder on poly(methyl methacrylate) (PMMA) substrate, from 2° to 30° at with 0.02° increment. Nitrogen sorption experiments were performed at 77 K on Micromeritics Instrument Corporation model 3Flex surface characterisation. The Brunauer-Emmett-Teller (BET) method was utilised to calculate the specific surface areas. By adopting the non-local density functional theory (NLDFT) model, the pore size distribution profile and pore volume were derived from the sorption curve. FT IR spectra were recorded on a Bruker Alpha with KBr disks. $^1$H NMR spectra were recorded on a Bruker 500 MHz NMR. Solid state $^{13}$C pulse magic angle spinning nuclear magnetic resonance spectroscopy ($^{13}$C CP/MAS NMR) spectra were recorded on a Bruker DNP-NMR 400 MHz. CHN Elemental analysis was recorded on a ThermoFisher Scientific FlashSmart CHNS Elemental Analyser by weighing the samples in tin containers and then combusted at high temperature with oxygen.

**Computational studies**. The monolayer structure of HFPTP-DMePDA-COF was constructed from the HFPTP and DMePDA building blocks using AuToGraFS and optimized using the self-consistent charge density functional tight binding (SCC-DFTB) method and the mio-0-1 parameter set and Lennard-Jones dispersion. All calculations were undertaken using DFTB+ version 1.3. Molecular modelling and Pawley refinement were carried out using a software package for crystal determination from PXRD pattern, implemented in Material Studio modelling version 8 (Accerlrys Inc.). We performed Pawley refinement to optimize the lattice parameters iteratively until the $R_{wp}$ and $R_p$ values converge. The pseudo-Voigt profile function for whole profile fitting and asymmetry correction function as Berar-Baldinozzi were used during the refinement processes. The final unit cell parameters and refinement factors are $a = b = 25.8092$ Å, $c = 4.5279$ Å, $\alpha = \beta = 90°$, $\gamma = 120°$, $R_{wp} = 3.72\%$ and $R_p = 2.15\%$, respectively.

**Vapour sorption**. Vapour sorption experiment was performed on a Quantachrome Instruments model iQ3 equipped with manifold, vapour source and circulating water bath. Samples were degassed under vacuum at 120 °C for 2 h prior to measurement. The water adsorption isotherms are performed at 283.15 K, 288.15 K and 298.15 K where the saturated vapour pressures were 9.2 torr, 12.8 torr and 23.8 torr respectively. The unit of measured volumetric vapour adsorption quantity was converted from cm$^3$ g$^{-1}$ to g g$^{-1}$ with an equation of w = (V/22414) × 18.020. The isosteric heat of adsorption ($Q_{st}$) was calculated with Clausius-Clapeyron equation

$$\triangle_{ads}H_w = R\left(\frac{\partial \ln p}{\partial\left(\frac{1}{T}\right)}\right)_W.$$

**Materials**. Dichloromethane, hexane and tetrahydrofuran (THF, stabilized with BHT) were purchased from Avantor Performance Materials. o-Dichlorobenzene (o-DCB), n-butanol (n-BuOH), dioxane, mesitylene, trifluoromethanesulfonic acid, paraformaldehyde, hydrobromic acid (41%), acetic acid and hydrazobenzene were purchased from TCI Chemicals. Toluene, methanol (MeOH) and dimethylformamide (DMF) were purchased from VWR Chemicals. Potassium carbonate and potassium permanganate were purchased from Goodrich Chemical Enterprise. 1,3,6,8-Tetrabromopyrene, tetrakis(triphenylphosphine)palladium(0), 1,4-dimethoxybenzene, 4-aminobenzonitrile, 2,3,6,7,10,11-hexabromotriphenylene, 4-formylphenylboronic acid, 1,4-phenylenediamine (PDA), 2,5-dimethyl-1,4-phenylenediamine (DMePDA), 1,3,5-benzenetricarboxylic acid and N-hydroxysuccinimide (NBS) were purchased from BLDPharm. Carbazole, lithium aluminum hydride, pyridinium chlorochromate and n-butyllithium solution (2.0 M in cyclohexane) were purchased from Sigma Aldrich.

**Synthesis**. 2,3,6,7,10,11-Hexakis(4-formylphenyl) triphenylene (HFPTP)[15], 4,4',4''-(1,3,5-triazine-2-4-6-triyl)trianiline (TTA)[30], 1,3,5-triformylbenzene (TFB)[31], 3,3',6,6'-tetraformyl-9,9'-bicarbazole (TFBCz)[19], 1,3,6,8-tetrakis(4-fomylphenyl) pyrene (TFPPy)[21], 1,1'-biphenyl-4,4'-diamine (BPDA)[32] and 2,5-dimethoxyterephthalaldehyde (DMTP)[33,34] were synthesized according to reported literatures.

**HFPTP-PDA-COF**[14]. An o-DCB/n-BuOH/2 M AcOH (4/6/1 by vol.; 1.1 mL) mixture of HFPTP (20 mg, 0.028 mmol) and PDA (14 mg, 0.129 mmol) in a Pyrex tube (10 mL) was degassed by three freeze-pump-thaw cycles. The tube was sealed and heated at 120 °C under vacuum for 7 days. The precipitate was collected by suction filtration and washed five times with THF and then Soxhlet with THF overnight. The powder was dried at 120 °C under vacuum overnight to yield HFPTP-PDA-COF in an isolated yield of 85%.

**HFPTP-DMePDA-COF**. An o-DCB/n-BuOH/12 M AcOH (5/5/1 by vol.; 1.1 mL) mixture of HFPTP (15 mg, 0.021 mmol) and DMePDA (9.3 mg, 0.068 mmol) in a Pyrex tube (10 mL) was degassed by three freeze-pump-thaw cycles. The tube was sealed and heated at 120 °C under vacuum for 7 days. The precipitate was collected by suction filtration and washed five times with THF and then Soxhlet with THF overnight. The powder was dried at 120 °C under vacuum overnight to yield HFPTP-DMePDA-COF in an isolated yield of 78%.

**HFPTP-BPDA-COF**[15]. An o-DCB/n-BuOH/6 M AcOH (5/5/1 by vol.; 1.1 mL) mixture of HFPTP (20 mg, 0.028 mmol) and BPDA (15.6 mg, 0.084 mmol) in a Pyrex tube (10 mL) was degassed by three freeze-pump-thaw cycles. The tube was sealed and heated at 120 °C under vacuum for 7 days. The precipitate was collected by suction filtration and washed five times with THF and then Soxhlet with THF overnight. The powder was dried at 120 °C under vacuum overnight to yield HFPTP-BPDA-COF in an isolated yield of 77%.

**TFBCz-PDA-COF**[19]. A 1,4-dioxane/mesitylene/6 M AcOH (5/5/1 by vol.; 1.1 mL) mixture of TFBCz (37 mg, 0.083 mmol) and PDA (18 mg, 0.17 mmol) in a Pyrex tube (10 mL) was degassed by three freeze-pump-thaw cycles. The tube was sealed and heated at 120 °C under vacuum for 3 days. The precipitate was collected by suction filtration and washed five times with THF and then Soxhlet with THF overnight. The powder was dried at 120 °C under vacuum overnight to yield TFBCz-PDA-COF in an isolated yield of 78%.

**TFPPy-PDA-COF**[21]. A 1,4-dioxane/6 M AcOH (10/1 by vol.; 1.1 mL) mixture of TFPPy (20 mg, 0.032 mmol) and PDA (7.7 mg, 0.071 mmol) in a Pyrex tube (10 mL) was degassed by three freeze-pump-thaw cycles. The tube was sealed and heated at 120 °C under vacuum for 3 days. The precipitate was collected by suction filtration and washed five times with THF and then Soxhlet with THF overnight. The powder was dried at 120 °C under vacuum overnight to yield TFPPy-PDA-COF in an isolated yield of 86%.

**TTA-TFB-COF**[20]. An o-DCB/n-BuOH/6 M AcOH (5/5/1 by vol.; 1.1 mL) mixture of TTA (35.4 mg, 0.1 mmol) and TFB (16.2 mg, 0.1 mmol) in a Pyrex tube (10 mL) was degassed by three freeze-pump-thaw cycles. The tube was sealed and heated at 120 °C under vacuum for 3 days. The precipitate was collected by suction filtration and washed five times with THF and then Soxhlet with THF overnight. The powder was dried at 120 °C under vacuum overnight to yield TTA-TFB-COF in an isolated yield of 84%.

**TPB-DMTP-COF**[22]. An o-DCB/n-BuOH/6 M AcOH (5/5/1 by vol.; 1.1 mL) mixture of TPB (30 mg, 0.076 mmol) and DMTP (20.5 mg, 0.15 mmol) in a Pyrex tube (10 mL) was degassed by three freeze-pump-thaw cycles. The tube was sealed and heated at 120 °C under vacuum for 3 days. The precipitate was collected by suction filtration and washed five times with THF and then Soxhlet with THF overnight. The powder was dried at 120 °C under vacuum overnight to yield TPB-DMTP-COF in an isolated yield of 81%.

**TTA-TFB-CMP**. A MeCN/6 M AcOH (10/1 by vol.; 1.1 mL) mixture of TTA (35.4 mg, 0.1 mmol) and TFB (16.2 mg, 0.1 mmol) in a Pyrex tube (10 mL) was degassed by three freeze-pump-thaw cycles. The tube was sealed and heated at 90 °C under vacuum for 1 day. The precipitate was collected by suction filtration and washed five times with THF and then Soxhlet with THF overnight. The powder was dried at 120 °C under vacuum overnight to yield TTA-TFB-CMP in an isolated yield of 82%.

## Data availability

The datasets generated during and/or analysed during the current study are available from the corresponding authors on request.

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

## Acknowledgements

D.J. acknowledges supports by the Singapore MOE tier 1 grant (R-143-000-C06-114), MOE tier 1 grant (R-143-000-C05-114) and MOE tier 2 grant (MOE-T2EP10220-0004).

## Author contributions

D.J. conceived the project, designed experiments and provided funds. K.T. S.T. and N.H. conducted the experiments. D.J. and K.T. wrote the manuscript and discussed the results with all authors. All data are reported in the main text and supplementary materials.

## Competing interests

The authors declare no competing interests.
