## [Peer Review File · Nature Communications]

REVIEWER COMMENTS

Reviewer #1 (Remarks to the Author):

The authors have reported an interesting study of water adsorption behavior in several covalent-organic frameworks (COFs). I particularly liked the idea of being able to create porous materials with 1D channels that have strips of hydrophilicity. I think such behavior is rather unique to COFs. Otherwise, water adsorption behavior as a function of hydrophobicity of pores of varying sizes is not new. The authors have neglected much relevant research over the past several decades in adsorption science. Water adsorption in mesoporous silicas are particularly relevant here as many of them present 1D channels as well and exhibit similar water isotherms. The authors should compare and contrast their results with the well-known adsorption literature.

A few additional comments/questions should be addressed:

1. The DMePDA COF saturates at much lower loading than the pore volume would indicate should be possible. The explanation of this discrepancy wasn't detailed enough. The authors should provide more evidence of their reasoning here.

2. The authors keep referring to "supermicropores" but the pore sizes do not meet the IUPAC definition. Pore width of 0.7 nm - 2 nm are micropores. Supermicropores refer to pore widths that are < 0.7 nm. The authors should be careful to not create new definitions and adhere to the IUPAC classifications. While COFs are relatively new, adsorption science is not. [Thommes et al., Pure Appl Chem, 2015: DOI 10.1515/pac-2014-1117]

3. The heat of adsorption appears to have been calculated using two data points. Applying the Clausius-Clapeyron equation requires at least 3 points for better accuracy (2 points will always make a perfect line).

Reviewer #2 (Remarks to the Author):

Jiang et al investigate the impact of COFs' design on water uptake and release by varying the shape, size, and chemical nature of the pores. This work is very interesting and could lead to improved materials to address this topical issue of water harvesting using porous materials. The work is very detailed and gives insights into important parameters needed for effective water uptake which is still poorly understood especially for COFs. I would recommend this exciting work for publication in Nature Communications after addressing the following minor issues:

1. Giving the importance of pore functionalization on reversible water uptake, the author should report elemental analysis results for all reported COFs after activation under an inert atmosphere. COFs are known to have structural defects which would add hydrophilic functionality (unreacted -NH₂) to the framework and thus enhance water uptake.
2. Full range IR spectrum should be included in the supporting information to see if there are unreacted -NH₂ groups
3. The reported pore size distribution graphs (figure 2) indicate that most COFs have pores in the micro and mesoporous range while the COFs in Figure 1 (top 2) have only micropores! This is confusing and it should be discussed briefly in the manuscript.
4. The authors mention that an increase by only 0.1 nm has a great impact on water uptake isotherms so how accurate is the method used for pore size distribution? According to Figure 2 (d,f, h), there are some mesopores in all COFs.
5. In my opinion, the title should be revised because the term “hydrophobic” is not a very accurate description of the investigated COFs in this study. It is true that most of the interior pore surface is made of C-H bonds, however, the imine-linkage is polar and has a great effect on water uptake as accepted by the authors. This is also the case of most MOFs which are made of aromatic rings and linked by polar secondary building units. A true hydrophobic COF

Point-to-Point Answers to Reviewers' Comments

Reviewer #1 (Remarks to the Author):

The authors have reported an interesting study of water adsorption behavior in several covalent-organic frameworks (COFs). I particularly liked the idea of being able to create porous materials with 1D channels that have strips of hydrophilicity. I think such behavior is rather unique to COFs. Otherwise, water adsorption behavior as a function of hydrophobicity of pores of varying sizes is not new. The authors have neglected much relevant research over the past several decades in adsorption science. Water adsorption in mesoporous silicas are particularly relevant here as many of them present 1D channels as well and exhibit similar water isotherms. The authors should compare and contrast their results with the well-known adsorption literature.

We appreciate these suggestive comments.

We have added new references 17 and 18 on silicas “17 Pires, J., Pinto, M., Estella, J. & Echeverría, J. C. Characterization of the hydrophobicity of mesoporous silicas and clays with silica pillars by water adsorption and DRIFT. *J. Colloid. Interface Sci.* **317**, 206–213, doi:doi.org/10.1016/j.jcis.2007.09.035 (2008); and 18 Björklund, S. & Kocherbitov, V. Alcohols react with MCM-41 at room temperature and chemically modify mesoporous silica. *Sci. Rep.* **7**, 9960, doi:10.1038/s41598-017-10090-x (2017).”, to the line “The water vapour sorption isotherms of HFPTP-PDA-COF (Fig. 3 and b, red dots and curves) resemble a typical S-shaped type-V sorption curve, which is characteristic of hydrophobic materials.” to emphasise mesoporous silica.

We have further added new reference 24 and 25 on silicas “24 Inagaki, S., Fukushima, Y., Kuroda, K. & Kuroda, K. Adsorption isotherm of water vapor and its large hysteresis on highly ordered mesoporous silica. *J. Colloid. Interface Sci.* **180**, 623–624, doi:doi.org/10.1006/jcis.1996.0345 (1996); and 25 Hwang, J., Kataoka, S., Endo, A. & Daiguji, H. Adsorption and desorption of water in two-dimensional hexagonal mesoporous silica with different pore dimensions. *J. Phys. Chem. C* **119**, 26171–26182, doi:10.1021/acs.jpcc.5b08564 (2015).”, to the line

“A further drawback is that the mesoporous COFs display a large hysteresis loop to complete the adsorption–desorption exchange cycle (Fig. 4i and ji, black and green dots and curves) similar to mesoporous silica FSM-16, MPS-1, MPS-2, and MPS-3.” to describe mesoporous silica.

A few additional comments/questions should be addressed:

1. The DMePDA COF saturates at much lower loading than the pore volume would indicate should be possible. The explanation of this discrepancy wasn't detailed enough. The authors should provide more evidence of their reasoning here.

We appreciate the comment.

We attributes the lower loading of HFPTP-DMePDA-COF to the steric obstruction by methyl units at linker sites. The methyl substitution positioned right next to the hydrophilic N-strips precludes the possibility of water cluster nucleation. We have added new sentences in details to the Results session for a better understanding.

“The much lower pore accessibility originates from the methyl units on the wall surface which impose a severe steric hindrance for water access to the neighbouring hydrophilic N chains. This spatial restriction extends along the z direction on each pore walls and causes difficulty of water cluster formation to reach a high pore filling.”

We have also highlighted a part in the Discussion session to explain the distinct vapour uptake behaviour of HFPTP-DMePDA-COF, as shown below. “In contrast, HFPTP-DMePDA-COF is different in the confinement process and mechanism. The methyl groups owing to hyperconjugation effect weaken polarisation of the C=N linkages¹ to decrease the ‘hydrophilicity’ and spatially set a steric barrier to prevent the access to water molecules (Fig. 3g). These two-fold effects broaden the induction pressure zone and eliminate capillary condensation. In this case, water molecules are confined by sluggish diffusion *via* water-water intermolecular hydrogen bonds, resulting in an isotherm convex to the pressure axis.”

2. The authors keep referring to "supermicropores" but the pore sizes do not meet the IUPAC definition. Pore width of 0.7 nm - 2 nm are micropores. Supermicropores refer to pore widths that are < 0.7 nm. The authors should be careful to not create new definitions and adhere to the IUPAC classifications. While COFs are relatively new, adsorption science is not. [Thommes et al., Pure Appl Chem, 2015: DOI 10.1515/pac-2014-1117]

We appreciate the comment.

We refer to the literatures^{2,3} in which pore < 0.7 nm is classified as ultramicropore while pore width of 0.7 nm – 2 nm is supermicropore. Trigonal HFPTP-PDA-COF, HFPTP-DMePDA-COF and HFPTP-BPDA-COF all have pore width of 0.7 nm – 2 nm, hence we defined them as supermicroporous material.

3. The heat of adsorption appears to have been calculated using two data points. Applying the Clausius-Clapeyron equation requires at least 3 points for better accuracy (2 points will always make a perfect line).

We appreciate this suggestive comment.

Accordingly, we have measured the vapour sorption at 15 °C and included the isotherms of all the COFs in Figures 3b and 4j. We re-evaluate the isosteric heat of adsorption with 3 points (10 °C, 15 °C and 25 °C) by applying Clausius-Clapeyron equation and summarise the results in Figures 3e and 4n. We have also replotted the isosteric heat of adsorption (Q_{st}) plots based on the 3 points analysis in Supplementary Fig. 7. The updated 3 point analysis results coincide with our findings previously.

We appreciate the reviewer very much for suggestive comments which improve the manuscript.

Reviewer #2 (Remarks to the Author):

Jiang et al investigate the impact of COFs' design on water uptake and release by varying the shape, size, and chemical nature of the pores. This work is very interesting can could lead to improved materials to address this topical issue of water harvesting using porous materials. The work is very detailed and gives insights into important parameters needed for effective water uptake which is still poorly understood especially for COFs. I would recommend this exciting work for publication in Nature Communications after addressing the following minor issues:

We appreciate these suggestive comments.

1. Giving the importance of pore functionalization on reversible water uptake, the author should report elemental analysis results for all reported COFs after activation under an inert atmosphere. COFs are known to have structural defects which would add hydrophilic functionality (unreacted -NH₂) to the framework and thus enhance water uptake.

We appreciate these suggestive comments.

Accordingly, we have measured the elemental analysis of all COFs and the findings are summarised in Supplementary Table 4. We have added a new sentence “ CHN Elemental analysis was recorded on a ThermoFisher Scientific FlashSmart CHNS Elemental Analyser by weighing the samples in tin containers and then combusted at high temperature with oxygen.” in the Methods session. The observed elemental composition coincides with the calculated composition by small discrepancy.

2. Full range IR spectrum should be included in the supporting information to see if there are unreacted -NH₂ groups.

We appreciate this comment.

We have added the full range FT IR spectra of all COFs to Supplementary Figs. 2, 11, and 12. We observed negligible N–H stretch signal at 3300 – 3500 cm⁻¹. This means that unreacted amine groups are not apparent in the COFs. Based on these observations, we believe that the high uptake stems from the regular channel for hydrophilic N strips access.

3. The reported pore size distribution graphs (figure 2) indicate that most COFs have pores in the micro and mesoporous range while the COFs in Figure 1 (top 2) have only micropores! This is confusing and it should be discussed briefly in the manuscript.

We appreciate this comment.

The minor mesopores are likely due to the structural defects which are commonly observed in trigonal COFs as reported.⁴ We have added new sentences “Minor large pores at 1.8, 4.1 and 3.8 nm are observed in HFPTP-PDA-COF, HFPTP-DMePDA-COF and HFPTP-BPDA-COF due to an incomplete connection between neighbouring pores.⁴ Nevertheless, these pores occupy a

negligible percentage in terms of pore volume and thus do not impact the water sorption behaviour.”, to the revised manuscript for a better understanding.

4. The authors mention that an increase by only 0.1 nm has a great impact on water uptake isotherms so how accurate is the method used for pore size distribution? According to Figure 2 (d,f, h), there are some mesopores in all COFs.

We appreciate this suggestive comment.

We deployed NLDFT model to derive the pore size distribution profile. This model is authentic and generally adopted for modelling the pore sizes of COFs. Literatures also support that this model is suitable for microporous COFs.⁴⁻⁶ As such, the pore size accuracy is reasonable to reveal 0.1 nm difference in HFPTP-PDA-COF and HFPTP-BPDA-COF.

The mesopores observed in COFs are attributed to an incomplete connection between neighbouring pores in trigonal COFs. These large pores occupy a very limited percentage in terms of pore volume and do not impact the overall water adsorption behaviour of COFs.

We have modified the sentences in the Methods session for a better understanding.

“By adopting the non-local density functional theory (NLDFT) model, the pore size distribution profile and pore volume were derived from the nitrogen sorption curve.”

5. In my opinion, the title should be revised because the term “hydrophobic” is not a very accurate description of the investigated COFs in this study. It is true that most of the interior pore surface is made of C-H bonds, however, the imine-linkage is polar and has a great effect on water uptake as accepted by the authors. This is also the case of most MOFs which are made of aromatic rings and linked by polar secondary building units. A true hydrophobic COF

We appreciate the comment.

Materials like metal organic frameworks, mesoporous silica, and porous carbon often exhibit water uptake zone larger than 0.3 relative pressure and are considered to be hydrophobic.⁷⁻¹⁰ Zeolites however, deliver steep water uptake lower than 0.1 relative pressure and is treated as hydrophilic.¹¹ We compared COFs with these material and consider to retain the term “hydrophobic” in the title as the uptake behaviour is more similar to the hydrophobic counterparts. The polar imine-linkage aids to shift the uptake zone to low pressure but the interaction is much weaker than ionic or hydrogen-bonding interactions in hydrophilic material. We think it is better to propose “pseudo-hydrophilicity strips” exist in these hydrophobic COFs.

We appreciate the reviewer very much for suggestive comments which improve the manuscript.

PS: We summarized the list of references below used in this point-to-point answer for your quick understandings.

References

- 1 Tao, S. *et al.* Confining H₃PO₄ network in covalent organic frameworks enables proton super flow. *Nat. Commun.* **11**, 1981, doi:10.1038/s41467-020-15918-1 (2020).
- 2 Sing, K. S. W. & Williams, R. T. Physisorption hysteresis loops and the characterization of nanoporous materials. *Adsorpt. Sci. Technol.* **22**, 773–782, doi:10.1260/0263617053499032 (2004).
- 3 Lastoskie, C., Gubbins, K. E. & Quirke, N. Pore size heterogeneity and the carbon slit pore: a density functional theory model. *Langmuir* **9**, 2693–2702, doi:10.1021/la00034a032 (1993).
- 4 Wang, P. *et al.* High-precision size recognition and separation in synthetic 1D nanochannels. *Angew. Chem. Int. Ed.* **58**, 15922–15927, doi:10.1002/anie.201909851 (2019).
- 5 Zhou, T.-Y., Xu, S.-Q., Wen, Q., Pang, Z.-F. & Zhao, X. One-step construction of two different kinds of pores in a 2D covalent organic framework. *J. Am. Chem. Soc.* **136**, 15885–15888, doi:10.1021/ja5092936 (2014).
- 6 Abuzeid, H. R., El-Mahdy, A. F. M. & Kuo, S.-W. Hydrogen bonding induces dual porous types with microporous and mesoporous covalent organic frameworks based on bicarbazole units. *Microporous Mesoporous Mater.* **300**, 110151, doi:doi.org/10.1016/j.micromeso.2020.110151 (2020).
- 7 Furukawa, H. *et al.* Water adsorption in porous metal–organic frameworks and related materials. *J. Am. Chem. Soc.* **136**, 4369–4381, doi:10.1021/ja500330a (2014).
- 8 Cmarik, G. E., Kim, M., Cohen, S. M. & Walton, K. S. Tuning the adsorption properties of UiO-66 via ligand functionalization. *Langmuir* **28**, 15606–15613, doi:10.1021/la3035352 (2012).
- 9 Inagaki, S., Fukushima, Y., Kuroda, K. & Kuroda, K. Adsorption isotherm of water vapor and its large hysteresis on highly ordered mesoporous silica. *J. Colloid. Interface Sci.* **180**, 623–624, doi:doi.org/10.1006/jcis.1996.0345 (1996).
- 10 Kaneko, K., Hanzawa, Y., Iiyama, T., Kanda, T. & Suzuki, T. Cluster-mediated water adsorption on carbon nanopores. *Adsorption* **5**, 7–13, doi:10.1023/A:1026471819039 (1999).
- 11 Dzhigit, O. M., Kiselev, A. V., Mikos, K. N., Muttik, G. G. & Rahmanova, T. A. Heats of adsorption of water vapour on X-zeolites containing Li⁺, Na⁺, K⁺, Rb⁺, and Cs⁺ cations. *J. Chem. Soc. Faraday Trans.* **67**, 458–467, doi:10.1039/TF9716700458 (1971).

REVIEWERS' COMMENTS

Reviewer #1 (Remarks to the Author):

The authors have adequately addressed the reviewer comments with a small exception. They somehow insist on using terms like "supermicropores" and reference literature from the 90s to justify the usage. I believe it is more appropriate to use terms that are approved by the IUPAC in 2015.

Reviewer #2 (Remarks to the Author):

I am very satisfied with the revised version of the manuscript; the authors did an excellent job addressing my comments and concerns. Very impressive work and I would recommend this manuscript for publication in Nature Communications as is.

Point-to-Point Answers to Reviewers' Comments

Reviewer #1 (Remarks to the Author):

The authors have adequately addressed the reviewer comments with a small exception. They somehow insist on using terms like "supermicropores" and reference literature from the 90s to justify the usage. I believe it is more appropriate to use terms that are approved by the IUPAC in 2015.

We appreciate these suggestive comments.

We rechecked the IUPAC naming in 2015¹ as suggested and we find that there might be some misunderstanding. The claim provided in the report "In the context of physisorption, it is expedient to classify pores according to their size (IUPAC recommendation, 1985): (i) pores with widths exceeding about 50 nm are called *macropores*; (ii) pores of widths between 2 nm and 50 nm are called *mesopores*; (iii) pores with widths not exceeding about 2 nm are called *micropores*."

Micropore filling may be regarded as a primary physisorption process. It is often useful to distinguish between the *narrow micropores* (also called *ultramicropores*) of approximate width < 0.7 nm and *wide micropores* (also called *supermicropores*)." describes that pore size smaller than 2 nm is classified as *micropore*, pore size smaller than 0.7 nm is classified as *ultramicropore* and those greater than 0.7 nm, is called *wide micropores* or *supermicropores*. The unclear description in the IUPAC technical report might have caused some misunderstanding. However, reading the claim twice, we conclude that supermicropores is suitable for our work. We hope to seek kind understanding from the reviewer that we are not redefining new terms and the definition is still being approved by IUPAC in 2015.

Reviewer #2 (Remarks to the Author):

I am very satisfied with the revised version of the manuscript; the authors did an excellent job addressing my comments and concerns. Very impressive work and I would recommend this manuscript for publication in Nature Communications as is.

We appreciate these suggestive comments.

References

- 1 Thommes, M. *et al.* Physisorption of gases, with special reference to the evaluation of surface area and pore size distribution (IUPAC Technical Report). *Pure Appl. Chem.* **87**, 1051–1069, doi:10.1515/pac-2014-1117 (2015).